# Efficient multi-fidelity computation of blood coagulation under flow

Manuel Guerrero-Hurtado[1], Manuel Garcia-Villalba[2], Alejandro Gonzalo[3], Pablo Martinez-Legazpi[4,5], Andrew M. Kahn[6], Elliot McVeigh[6,7,8], Javier Bermejo[5,9,10,11], Juan C. del Alamo[3,12,13], Oscar Flores[1]*

1 Department of Aerospace Engineering, Universidad Carlos III de Madrid, Leganés, Spain, 2 Institute of Fluid Mechanics and Heat Transfer, TU Wien, Vienna, Austria, 3 Department of Mechanical Engineering, University of Washington, Seattle, Washington, United States of America, 4 Department of Mathematical Physics and Fluids, Facultad de Ciencias, Universidad Nacional de Educación a Distancia, UNED, Spain, 5 CIBERCV, Madrid, Spain, 6 Division of Cardiovascular Medicine, University of California San Diego, La Jolla, California, United States of America, 7 Department of Bioengineering, University of California San Diego, La Jolla, California, United States of America, 8 Department of Radiology, University of California San Diego, La Jolla, California, United States of America, 9 Hospital General Universitario Gregorio Marañón, Madrid, Spain, 10 Instituto de Investigación Sanitaria Gregorio Marañón, Madrid, Spain, 11 Facultad de Medicina, Universidad Complutense de Madrid, Madrid, Spain, 12 Center for Cardiovascular Biology, University of Washington, Seattle, Washington, United States of America, 13 Division of Cardiology, University of Washington, Seattle, Washington, United States of America

* oflores@ing.uc3m.es

**Data Availability Statement:** All relevant data are within the paper, its Supporting information files, and on Zenodo (DDOI:10.5281/zenodo.8344615).

## Abstract

Clot formation is a crucial process that prevents bleeding, but can lead to severe disorders when imbalanced. This process is regulated by the coagulation cascade, a biochemical network that controls the enzyme thrombin, which converts soluble fibrinogen into the fibrin fibers that constitute clots. Coagulation cascade models are typically complex and involve dozens of partial differential equations (PDEs) representing various chemical species' transport, reaction kinetics, and diffusion. Solving these PDE systems computationally is challenging, due to their large size and multi-scale nature. We propose a multi-fidelity strategy to increase the efficiency of coagulation cascade simulations. Leveraging the slower dynamics of molecular diffusion, we transform the governing PDEs into ordinary differential equations (ODEs) representing the evolution of species concentrations versus blood residence time. We then Taylor-expand the ODE solution around the zero-diffusivity limit to obtain spatio-temporal maps of species concentrations in terms of the statistical moments of residence time, $\overline{t_R^p}$, and provide the governing PDEs for $\overline{t_R^p}$. This strategy replaces a high-fidelity system of $N$ PDEs representing the coagulation cascade of $N$ chemical species by $N$ ODEs and $p$ PDEs governing the residence time statistical moments. The multi-fidelity order ($p$) allows balancing accuracy and computational cost providing a speedup of over $N/p$ compared to high-fidelity models. Moreover, this cost becomes independent of the number of chemical species in the large computational meshes typical of the arterial and cardiac chamber simulations. Using a coagulation network with $N = 9$ and an idealized aneurysm geometry with a pulsatile flow as a benchmark, we demonstrate favorable accuracy for low-order models of $p = 1$ and $p = 2$. The thrombin concentration in these models departs from the high-fidelity solution by under 20% ($p = 1$) and 2% ($p = 2$) after 20 cardiac cycles. These multi-fidelity

**Funding:** MGH, MGV and OF have been partially supported by the Spanish Research Agency and the European Regional Development Fund, under grant number PID2019-107279RB-I00. MGH, MGV, PML, JB and OF have been partially supported by the Comunidad de Madrid and the European Regional Development Fund, under grant number Y2018/BIO-4858 PREFI-CM, and by the Instituto de Salud Carlos III and the European Regional Development Fund, under grant numbers PI15/02211-ISBITAMI and DTS/1900063-ISBIFLOW. AG, EMcV, AK and JCdA have been partially supported by the US National Institutes of Health, under grant 1R01HL160024. JCdA has been partially supported by the US National Insitutes of Health, under grant number 1R01HL158667. The funders had no role in study design, data collection and analysis, decision to publish, or preparation of the manuscript.

**Competing interests:** The authors have declared that no competing interests exist.

models could enable new coagulation analyses in complex flow scenarios and extensive reaction networks. Furthermore, it could be generalized to advance our understanding of other reacting systems affected by flow.

## Author summary

The coagulation cascade is an intricate biochemical process that prevents excessive bleeding while maintaining vascular integrity. Modeling this process involves dozens of interdependent chemical reactions with disparate kinetics. Moreover, the reacting species are transported by the flow, leading to complex spatio-temporal dynamics. Consequently, the computational cost of modeling the coagulation cascade in flowing blood prohibits realistic simulations. To overcome this challenge, we introduce a new multi-fidelity approach that exploits the slow diffusion of chemical species to decouple simulating their flow transport and chemical reaction. This approach achieves a significant reduction in computational requirements while maintaining the accuracy of our simulations. We anticipate this new multi-fidelity approach will make coagulation cascade simulations in physiologically relevant scenarios accessible to many researchers. This technique will also enable studies necessitating multiple parametric runs, such as sensitivity or uncertainty quantification analyses. This advancement is poised to benefit medical professionals and researchers, opening new horizons in our understanding of coagulation processes and the effects of blood thinners.

## Introduction

Blood coagulation, or clotting, is a highly regulated mechanism vital in sealing wounded blood vessels to prevent bleeding. Abnormal or excessive clotting can result in serious medical conditions, such as stroke and deep vein thrombosis. Therefore, blood coagulation has been the subject of extensive research, and understanding its mechanisms is crucial to diagnose and manage numerous diseases.

The initiation of blood coagulation is an enzymatic cascade that amplifies thrombin concentration in plasma, activating fibrin polymerization to form a clot [1, 2]. This cascade can be started via extrinsic and intrinsic pathways. The extrinsic pathway is triggered by a vessel-injury-mediated release of coagulation factor VII and tissue factor (TF) into the bloodstream. The intrinsic pathway is auto-initiated, i.e., it does not require exposure to an extravascular tissue factor, and begins with the activation of plasma factors XII, XI, IX, and VIII. Both pathways eventually activate factor X, converging into the common pathway that amplifies thrombin, which in turn converts fibrinogen into fibrin filaments and activates factor XIII, which cross-links the fibrin mesh [3, 4]. The initiation, propagation, and inhibition of this complex process involve a network of over 80 known biochemical reactions [5].

Since thrombosis is a ubiquitous complication of cardiovascular diseases and device implantation [6, 7], there is an abundance of computational models considering coagulation in diverse physiological and anatomical settings. These models may be coupled to computational fluid dynamics (CFD) solvers to capture the interactions between the abnormal blood flow, hypercoagulability, and vessel injury characteristic of thrombus formation. However, the problem is very complicated, and state-of-the-art studies including all interactions are limited. Intra-luminal thrombogenesis in an abdominal aortic aneurysm has been studied using an

18-equation model based on tissue factor activation [8, 9], providing an integrated mechano-chemical picture of the process [10]. The same model was used later to study thrombogenesis in an infarcted left ventricle [11]. More complex models have also been used in the literature, either increasing the number of reaction equations involved in the model [12], or including the effects of tissue factor, platelet activation, and clot porosity on thrombus growth [13]. Other models have analyzed the contribution of intrinsic and extrinsic pathways at different timescales, highlighting the multi-stage character of the coagulation process [14].

Eulerian-Lagrangian approximations can be used to integrate coagulation cascade reaction-advection-diffusion equations with platelet activation and deposition. Researchers have used this approach to describe the evolution of non-activated and activated platelet concentrations [15], using the simulated velocity fields to track platelet activation and accumulation. The equations were solved using a stabilized finite element method. The accumulation model [16] accounts for various factors, including plasma-phase and membrane-phase reactions, coagulation inhibitors, and the presence of activated and unactivated platelets. Other groups have used similar strategies, like coupling a calibrated platelet aggregation model, which accounts for adhesion forces between platelet-platelet and platelet-wall at low and high shear rate levels, with an extrinsic coagulation cascade initiation model [17]. In this case, the coagulation cascade was based on a model using 23 chemical species [18].

The large number of coupled partial differential equations (PDEs) representing the reaction, advection, and diffusion of the species involved in the coagulation cascade creates stringent requirements for numerical simulation. This problem is aggravated by the high numerical cost of solving each PDE, owing to the disparate timescales associated with the flow, reaction kinetics, and diffusion [19–21]. The dimensional parameters involved in these processes for mid-size arteries are: flow velocity, $U_c \sim 10$cm/s; vessel diameter, $L_c \sim 1$cm; the cardiac cycle's period, $t_c \approx 1s$; timescales of enzymatic reaction kinetics, ranging from a few seconds to hundreds of seconds ($t_r \sim 10^2$s) [9, 20, 22]; and mass diffusivity coefficients for species in blood, $D_i \sim 10^{-6}$cm$^2$/s [10, 23, 24]. The relatively slow reaction times in these systems require running simulations over many cardiac cycles to reach convergence, i.e., $t_r \gg t_c$. Moreover, the slow diffusion of reactive species creates extremely thin layers in their concentration fields since the Peclet number, $Pe = U_c L_c/D_i = t_d/t_a \approx 10^7$, which measures the ratio between convective and diffusive transport, is very large. For reference, the Schmidt number, which measures the ratio between the viscous diffusion acting in the Navier-Stokes equations and the mass diffusivity acting in the coagulation system's equations, is $Sc = \nu/D_i \sim 10^4$ where $\nu \approx 4 \times 10^{-2}$cm$^2$/s is the kinematic viscosity of blood. Consequently, the spatial discretization of the coagulation system's equations requires much finer computational grids than the ones used to solve the Navier-Stokes equations. This problem is well described in CFD literature and it is common to many other reactive and non-reactive cardiovascular transport problems [25].

Previous simulations of the coagulation cascade under flow have proposed concessions to reduce computational cost. First, the multi-scale nature of reaction kinetics have been simplified by assuming that fast-reacting species are in equilibrium, leading to reduced models with fewer chemical species [23], or by using phenomenological models [26]. The latter have been used to investigate hypercoagulability in the left heart by considering fibrin production from fibrinogen and thrombin [27]. Second, in most if not all studies, the Peclet number has been decreased explicitly by prescribing unphysically high values for $D_i$ [10, 11], or implicitly by using a diffusive numerical discretization (e.g., upwinding first-order finite differences). In non-reactive transport problems, this concession has been justified on the basis of accounting for additional noise sources and as long as the effective $Pe$ remains $\gg 1$ [28]. However, its adequacy is more questionable in reacting problems like the coagulation cascade, where the

reaction timescale is intermediate between the transport and diffusive timescales. Another compromise adopted by some authors is to shorten the time integration to a few cardiac cycles, focusing only on the initial phases of thrombin activation [11]. Finally, many studies have considered idealized two-dimensional geometries to save computational cost [10, 13, 20, 21, 27]. In summary, there is an unmet need for computationally efficient strategies to model the coagulation cascade under flow.

We introduce a multi-fidelity modeling approach to significantly reduce the computational cost of coagulation cascade simulations in flowing blood and make this cost independent of the number of chemical species. The multi-fidelity approach transforms the reaction-advection-diffusion equations for species concentrations ($u_i$, $i = 1, \ldots, N$) into a system of ODEs by using blood residence time ($\overline{t_R}$) as the independent variable. The resulting model requires integrating only one PDE for $\overline{t_R}$ and $N$ ODEs for the coagulatory species, whose concentration fields can be mapped as $u_i(\overline{t_R}(\mathbf{x}, t))$. The transformation is exact for zero diffusivity. For small, finite diffusivity, the model can be Taylor-expanded in terms of the residence time statistical moments, i.e., $\overline{t_R^2}, \overline{t_R^3}, \ldots \overline{t_R^p}$, to derive a family of customizable, multi-fidelity models that offer a balance between cost and accuracy. We compare 1st-order (MuFi-1: 1 PDE, $N$ ODEs) and 2nd-order (MuFi-2: 2 PDEs, $N$ ODEs) multi-fidelity models with the high-fidelity (HiFi: $N$-PDEs) model for pulsatile flow through an aneurysm-like geometry, using a 9-species coagulation system [23] as a benchmark. Overall, the MuFi-1 and HiFi models show good agreement up to $t \approx 10 t_c$ cardiac cycles, while this agreement is improved and extended to longer times ($t \approx 20 t_c$) for MuFi-2. The proposed family of multi-fidelity coagulation models could benefit researchers in the field by enabling them to simulate and analyze complex blood coagulation phenomena more quickly and accurately, thus advancing our understanding of the underlying mechanisms and informing clinical practice.

## Methods

This section presents a multi-fidelity model to reduce the cost of simulating coagulation networks of $N$ species in flowing blood. To facilitate the model's presentation, we first review the standard high-fidelity model (HiFi: $N$-PDE) and introduce its first-order (MuFi-1: 1-PDE, $N$-ODE) approximation, which neglects diffusion. We then introduce the second-order approximation (MuFi-2: 2-PDE, $N$-ODE) accounting for small, finite diffusion, define our benchmark flow problem and coagulation reaction system, and describe the numerical discretization methods.

### High-fidelity and first-order multi-fidelity coagulation models

We consider blood as a continuum in space and time ($\mathbf{x}$, $t$) flowing with velocity $\mathbf{v}(\mathbf{x}, t)$, and model its coagulation by the system of reaction-advection-diffusion equations

$$\frac{Du_i}{Dt} = \frac{\partial u_i}{\partial t} + \mathbf{v} \cdot \nabla u_i = R_i + D_i \nabla^2 u_i, \qquad \text{for } i = 1, \ldots, N, \qquad (1)$$

where $D/Dt$ denotes material derivative, the subindex $i$ indicates each of the $N$ species involved in coagulation system, $u_i(\mathbf{x}, t)$ its concentration field, $R_i(u_1, u_2, \ldots, u_N)$ its reaction rate from chemical kinetics, and $D_i$ its diffusivity coefficient. This system of $N$ PDEs is denoted the high-fidelity (HiFi) model. Assuming that $\mathbf{v}(\mathbf{x}, t)$ is known, this HiFi model can be solved with some appropriate initial and boundary conditions for $u_i$. For simplicity, we consider uniform initial conditions $u_i(\mathbf{x}, 0) = u_{i,0}$, Dirichlet boundary conditions at the domain flow inlets ($u_i = u_{i,0}$), and homogeneous Neumann boundary conditions ($\partial u_i / \partial n = 0$) at solid surfaces and flow outlets.

 

The Eq (1) can be written in non-dimensional form using the flow velocity scale $U_c$ and vessel length scale $L_c$,

$$\frac{Du_i}{D\tau} = Da\tilde{R}_i + \frac{1}{Pe}\nabla^2 u_i, \tag{2}$$

where $\tau = tU_c/L_c$ is a dimensionless time variable, $\tilde{R}_i = t_r R_i$ is a dimensionless reaction rate, the Damköhler number $Da = L_c/(t_r U_c)$ measures the relative importance of reaction kinetics (rate $t_r^{-1}$) and convective terms, and the Péclet number $Pe = U_c L_c/D_i$ measures the relative importance of convection over diffusion. Using typical values corresponding to mid-size arteries and the reaction rate and diffusivity of coagulation cascade species (i.e. $U_c \sim 10$ cm/s, $L_c \sim 1$ cm, $t_r \sim 10^2$ s, $D_i \sim 10^{-6}$ cm$^2$/s) yields $Da \sim 10^{-3}$ and $1/Pe \sim 10^{-7}$, so both terms on the right-hand side of Eq (2) are small. However, the reaction term is the only forcing in the equation and cannot be neglected, while the diffusive term is even smaller and can be neglected. Doing so simplifies Eq (1) to

$$\frac{Dg_i}{Dt} = R_i, \tag{3}$$

where $g_i$ approximates $u_i$ in the limit of zero molecular diffusivity. The simplified model has no mixing [29], making the reaction rate within each fluid element independent of the species concentrations in surrounding elements, and exclusively dependent on its *age*. Consequently, it should be possible to write an ODE system for the coagulation system of each fluid element in a Lagrangian frame that follows the element as it moves with the flow. To avoid the complication of tracking Lagrangian trajectories, we follow previous works [30–33] and resort to the PDE governing the residence time

$$\frac{D\overline{t_R}}{Dt} = 1, \tag{4}$$

which can be used to calculate the *age* of fluid elements in the region of interest. Applying the chain rule on Eq (3) and taking into account Eq (4), we obtain

$$\frac{Dg_i}{Dt} = \frac{dg_i}{d\overline{t_R}}\frac{D\overline{t_R}}{Dt} = \frac{dg_i}{d\overline{t_R}}, \tag{5}$$

which simplifies the HiFi PDE system Eq (1) into the ODE system

$$\frac{dg_i}{d\overline{t_R}} = R_i(g_1, g_2, \ldots, g_N). \tag{6}$$

We note that this ODE system is the same for all fluid elements, as it has no explicit dependence on **x**, and each fluid element's pathline information is implicitly encoded by the spatial dependence of $\overline{t_R}(\mathbf{x}, t)$. This model involves solving a system of $N$ ODEs (i.e., Eq (6) for $i = 1 \ldots N$) to obtain $g_i(\overline{t_R})$, solving one PDE to calculate $\overline{t_R}$, and mapping $u_i(\mathbf{x}, t) \approx g_i(\overline{t_R}(\mathbf{x}, t))$. Because it combines solving ODEs and PDEs, we consider this approximate model a multi-fidelity (MuFi) model. When $N$ is large, the MuFi model can be significantly cheaper to run than the HiFi model, which involves solving $N$ PDEs.

## Higher-order multi-fidelity approximations

In the previous section, we used overline notation for residence time to emphasize that $\overline{t_R}$ as defined in Eq (4) is the *ensemble* average age of all molecules within a fluid element, defined as

 

the first integral moment of its probability density function, $f_T$:

$$\overline{t_R}(\mathbf{x}, t) = \int_{-\infty}^{\infty} T f_T(T; \mathbf{x}, t) \, dT. \tag{7}$$

Considering residence time as a stochastic variable is important because, as discussed in the Introduction, the numerical solutions to transport PDEs like Eq (4) explicitly include unphysically large diffusivities or employ discretization methods that implicitly introduce numerical diffusivity. Numerical dissipation helps control instabilities and spurious oscillations but it is a source of numerical error [34]. This error makes $\overline{t_R}$ satisfy an equivalent differential equation (EDE) instead of the theoretical PDE given by Eq (4). The specific form of the EDE depends on the details of the temporal integration scheme and the approximation used to discretize the spatial derivatives. For commonly used, first-order, dissipative methods, the EDE for $\overline{t_R}$ takes the form

$$\frac{D\overline{t_R}}{Dt} = 1 + D_n \nabla^2 \overline{t_R}, \tag{8}$$

where the coefficient $D_n \sim c\Delta x/2$ represents the diffusivity of the numerical method, where $c$ is the flow characteristic velocity and $\Delta x$ the mesh spatial resolution [34].

In this *effective* scenario, diffusivity creates uncertainty in the residence time. This phenomenon can be shown using Itô's differentiation [35] to derive the EDE for the second-order moment of the residence time, $\overline{t_R^2}$, as

$$\frac{D\overline{t_R^2}}{Dt} = 2\overline{t_R} + D_n \nabla^2 \overline{t_R^2}, \tag{9}$$

where

$$\overline{t_R^2}(\mathbf{x}, t) = \int_{-\infty}^{\infty} T^2 f_T(T; \mathbf{x}, t) dT. \tag{10}$$

Then, it is possible to express this equation in terms of the residence time variance, $\sigma_T^2 = \overline{t_R^2} - \overline{t_R}^2$, as

$$\frac{D\sigma_T^2}{Dt} = 2D_n |\nabla \overline{t_R}|^2 + D_n \nabla^2 \sigma_T^2. \tag{11}$$

The interested reader can find the derivation of Eqs (8) and (11) in S1 Appendix. Of note, when $D_n = 0$ and $f_T$ is a Dirac delta function, Eqs (8) and (9) yield $\overline{t_R^2} = \overline{t_R}^2$ and zero residence time variance, i.e., $\sigma_T^2 = 0$. But when $D_n \neq 0$, the non-negative forcing term in Eq (11) causes $\sigma_T^2$ to increase unless residence time is constant. Note also that for higher order numerical methods, the differential operator multiplying $D_n$ in the EDEs (8) and (9) will involve higher order derivatives, with a dispersive or dissipative character depending on the order of the temporal and spatial discretizations. This will result in more complex evolution equations for $\sigma_T^2$, without changing the fact that numerical diffusion results in an increase on $\sigma_T^2$ in regions with strong gradients of $\overline{t_R}$.

The growth of $\sigma_T^2$ causes errors in the MuFi model derived in the previous section as time increases. To illustrate these errors and derive higher-order corrections, it is convenient to express the concentration of chemical species as

$$u_i(\mathbf{x}, t) = \int_{-\infty}^{\infty} U_i f_{U_i}(U_i; \mathbf{x}, t) \, dU_i = \int_{-\infty}^{\infty} g_i(T) f_T(T; \mathbf{x}, t) dT, \tag{12}$$

where $U_i$ is the concentration's statistical variable, $f_{U_i}(U_i; \mathbf{x}, t)$ is its probability density function, and $f_T(T; \mathbf{x}, t)$ is the probability density function of the residence time. The substitution $U_i = g_i(T)$ is warranted because, in the absence of diffusion, $\sigma_T$ is zero and $f_T$ is a Dirac delta, yielding $u_i = g_i(\overline{t_R})$, consistent with the definition of $g_i$ in Eq (3).

Assuming next that $g_i(T)$ is second-order differentiable, we can Taylor-expand it around $T = \overline{t_R}$ and integrate the expansion to obtain

$$u_i(\mathbf{x}, t) \approx g_i(\overline{t_R}) + g_i''(\overline{t_R}) \frac{\sigma_T^2}{2}, \tag{13}$$

where the primes denote derivatives. This result demonstrates that the MuFi model $u_i = g_i(\overline{t_R})$ error grows with the residence time variance for non-linear reaction systems (i.e., those with $g'' \neq 0$). But, more important, it provides a high-order correction that is also an inexpensive MuFi model as it only requires solving one additional PDE.

In summary, we introduce two multi-fidelity models that balance cost with accuracy:

- **First-order (MuFi-1):**

  - Solve N ODEs Eq (6) to calculate $g_i(t)$.

  - Solve one PDE Eq (4) to calculate $\overline{t_R}(\mathbf{x}, t)$.

  - Map $u_i(\mathbf{x}, t) \approx g_i[\overline{t_R}(\mathbf{x}, t)]$.

- **Second-order (MuFi-2):**

  - Solve N ODEs Eq (6) to calculate $g_i(t)$ and its second temporal derivative $g_i''(t)$.

  - Solve two PDEs: Calculate $\overline{t_R}(\mathbf{x}, t)$ from Eq (4) and $\overline{t_R^2}(\mathbf{x}, t)$ from

$$\frac{D\overline{t_R^2}}{Dt} = 2\overline{t_R}, \tag{14}$$

  which follows from ignoring the numerical diffusion term from the EDE in Eq (9).

  - Calculate $\sigma_T^2 = \overline{t_R^2} - \overline{t_R}^2$, and map $u_i(\mathbf{x}, t) \approx g_i[\overline{t_R}(\mathbf{x}, t)] + g_i''[\overline{t_R}(\mathbf{x}, t)]\sigma_T^2(\mathbf{x}, t)/2$.

This procedure can be extended to higher order approximations by retaining additional terms in the Taylor expansion of $g_i$ around $\overline{t_R}$, and solving additional PDEs for higher-order moments of the residence time. For example, the MuFi model of third order (involving N-ODEs and three PDEs) is derived in S2 Appendix. Finally, we note that the PDEs to be solved in the MuFi-2 model are the *true* PDEs (i.e., Eqs (4) and (14)), not the EDEs (Eqs (8) and (9)). If the PDEs explicitly include diffusivity, then a diffusive term should be added. If not, the resulting $\sigma_T^2$ will capture the effect of the discretization's numerical diffusivity, $D_n$, regardless of the order of the numerical method employed to solve Eqs (4) and (14). We emphasize that explicit knowledge of $D_n$ is not required, which is advantageous since this diffusivity may be constant or vary in space and time depending on the numerical discretization.

## Computational cost estimates

We estimate the computational cost of running a HiFi model in $D$ dimensions to be $\varphi N n^{D+1}$ floating point operations (FLOPs), where $N$ is the number of species, $\varphi \sim O(10^2)$ is a parameter that depends on the numerical discretization scheme and the function evaluations needed to calculate the reaction rates, and $n$ is the number of elements in the spatial mesh along each direction. Note that the exponent $D + 1$ reflects that the temporal resolution is linked to the

spatial resolution by the CFL condition to guarantee an accurate temporal integration. As a consequence, the number of time steps required to reach a finite integration time is proportional to $n$. On the other hand, a MuFi-p model is estimated to require $\beta Nn + \theta pn^{D+1}$ FLOPs, where the first term is the cost of integrating the $N$ ODEs for the species reaction kinetics and the second term is the cost of solving the $p$ PDEs governing the statistical moments of residence time. Like in the HiFi model, the parameters $\beta$ and $\theta$ depend on the numerical implementation and the right-hand-side terms in the governing equations. It is worth noting that $\theta < \varphi$ because the forcing terms in the residence time equations (see e.g., Eqs 4 and 14) are simpler than the reaction rate terms in the equations governing species concentration (see S4 Appendix).

Based on these estimates and assuming that $n$ is a large number as in, e.g., spatially resolved simulations of flow through arteries or the cardiac chambers, we make two remarks. First, we note that the cost of MuFi models becomes effectively independent of the number of species, $N$, since $D \geq 1$. Second, we note that MuFi models achieve a speedup $\sim (\varphi/\theta)(N/p) > N/p$. Furthermore, MuFi models significantly reduce the memory allocation necessary to run simulations, which could lead to additional speedups in parallel implementations by avoiding the overheads associated with message passing and loss of cache memory coherence.

## Test case: Coagulation cascade in an idealized aneurysm

To compare the multi-fidelity models MuFi-1 and MuFi-2 with the high-fidelity model based on the PDE system (1), we considered a simplified coagulation cascade model under pulsatile flow through an idealized two-dimensional geometry (Fig 1). This flow geometry broadly resembles a cerebral aneurysm or the left atrial appendage, two cardiovascular sites associated with thrombosis [36–40]. The parent vessel is modeled as a straight tube of diameter $H$, and the aneurysm is modeled as a circular cavity of radius 0.75$H$. The center of the cavity is located such that the aneurysm neck size is $H$. The corners at the aneurysm neck are smoothed with a radius of curvature equal to 0.067$H$, to avoid sharp corners in the geometry. The pulsatile flow was driven by imposing a two-dimensional Womersley flow as the inflow boundary condition (see S3 Appendix). The

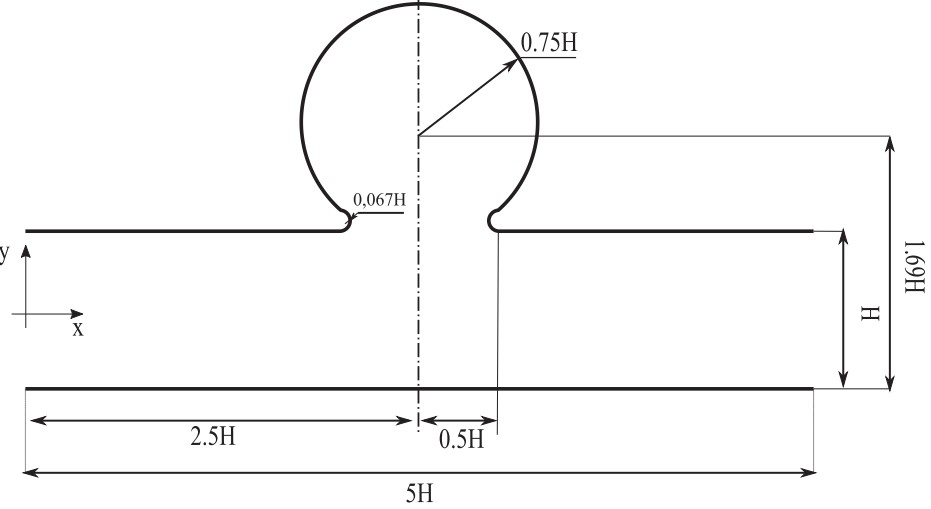

**Fig 1. Geometry.** Idealized aneurysm geometry.

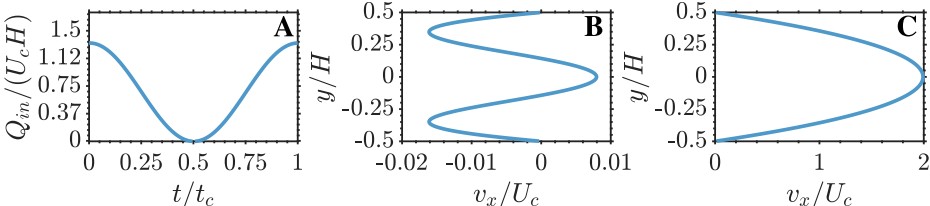

**Fig 2. Womersley flow profiles.** A: Time evolution of the mass flow rate through the vessel. B and C: Velocity profile at the inlet at $t = 0.5 t_c$ and $t = t_c$.

Reynolds and Womersley numbers are $Re = U_c H/\nu = 500$ and $\alpha = \sqrt{2\pi H^2/(t_c \nu)} = 10$, respectively, where $U_c$ is the maximum velocity. These values of $Re$ and $\alpha$ are representative of intracranial saccular aneurysms [41]. Fig 2 shows the time history of the mass flow rate and the inlet velocity profiles at two time instants, corresponding to the minimum ($t/t_c = 0.5$) and maximum ($t/t_c = 1$) flow rates through the vessel. While the waveform of $Q(t)$ does not include all the temporal complexity of physiological waveforms, it does include the acceleration and deceleration phases needed to drive the flow in the cavity, as described in section Results.

To model the coagulation cascade, we chose a 9-species system considering prothrombin (II), thrombin (IIa), fibrin (Ia), PCa, and factors XIa, IXa, Xa, VIIIa, and Va [23]. The corresponding source terms in Eq (1) are detailed in S4 Appendix. The equations describe the activation of prothrombin (II) into thrombin (IIa) by factors Va and Xa. In turn, thrombin activates the production of factors Va, VIIIa, XIa, and the system's main inhibitor, PCa. The reaction rates for the model are adopted from previous works [23, 42] and are reported in Table A in S4 Appendix.

Fig 3 illustrates the evolution of the 9 species in stagnant blood (i.e., $\mathbf{v} = \mathbf{0}$) for uniform initial concentrations (see Table 1), chosen within physiologically plausible ranges to ensure substantial thrombin growth within 20 cardiac cycles. This timescale aligns with the peak residence time values observed in the left atrial appendage (LAA) [43, 44]. These conditions lead to an accumulation of thrombin and factors Xa and VIIIa over the initial 10–17 cardiac cycles, followed by a rapid decrease in thrombin concentration over the subsequent 15 cycles.

## Numerical methods

We used the in-house code TUCAN [45, 46] to solve the Navier-Stokes equations for Newtonian, incompressible flow in the configuration described in the previous section. The

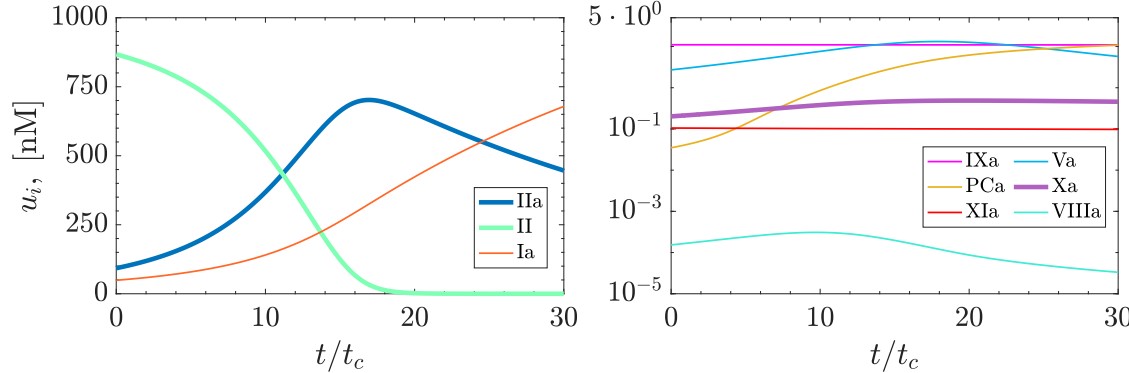

**Fig 3. Evolution of coagulation cascade species.** Obtained by solving Eq (6) for the 9-species coagulation model.

**Table 1. Initial conditions.**

| Species Concentrations | Initial condition [nM] |
|---|---|
| $u_{XIa}$ | 0.105 |
| $u_{IXa}$ | 11.024 |
| $u_{Xa}$ | 0.202 |
| $u_{IIa}$ | 92.626 |
| $u_{II}$ | 867.564 |
| $u_{VIIIa}$ | $1.534 \cdot 10^{-4}$ |
| $u_{Va}$ | 2.713 |
| $u_{PCa}$ | $3.488 \cdot 10^{-2}$ |
| $u_{Ia}$ | 48.811 |

numerical discretization used second-order finite differences on a Cartesian staggered grid. The temporal integration was performed with a three-stage, low-storage, semi-implicit Runge-Kutta scheme. The no-slip boundary condition at the vessel walls was modeled by the immersed boundary method [47]. After discarding initial transient effects, the velocity field (**v**(**x**, $t$)) computed by TUCAN was sampled at constant time intervals ($t_{samp} = t_{cycle}/35$) and stored to be linearly interpolated for integrating Eqs (1), (4) and (14). To assess the convergence of the velocity field, we performed a grid refinement study employing four resolutions, $\Delta x/H = 1/38$, $\Delta x/H = 1/75$, $\Delta x/H = 1/150$ and $\Delta x/H = 1/300$. Each simulation was run with a constant time step $\Delta t$ that ensured the Courant number to be $CFL = \max(|u(\mathbf{x})|)\Delta t/\Delta x \approx 0.1$. The relative error was defined with respect to the case with $\Delta x/H = 1/300$,

$$\varepsilon_k = \frac{|\overline{\omega}_k - \overline{\omega}_{300}|}{\overline{\omega}_{300}} \quad , \tag{15}$$

where

$$\overline{\omega}_k(t) = \frac{1}{\Omega_{cav}} \int \int_{\Omega_{cav}} |\omega(x, y, t)| d\Omega$$

is the averaged absolute value of the vorticity in the cavity computed with a spatial resolution $\Delta x = H/k$, and $\Omega_{cav}$ is the volume of the cavity. Table 2 displays the values of the relative error for each resolution at three time points ($t/t_c = 0$, $t/t_c = 0.33$, and $t/t_c = 0.67$). The case with resolution $\Delta x/H = 1/150$ was selected for the present study because it yielded a relative error consistently below 1% throughout the cardiac cycle.

A third-order weighted essentially non-oscillatory (WENO) scheme [48] was implemented to integrate the advection-reaction-diffusion PDEs (Eqs (1), (4) and (14)) as in previous works [43, 44]. This scheme locally adjusts numerical diffusivity to damp convective fluxes perpendicular to sharp scalar fronts, preventing spurious oscillations while at the same time keeping the overall numerical diffusivity low. The systems of PDEs (1), (4) and (14), and the system of

**Table 2. Grid refinement study.**

| Relative Error | $t/t_c = 0$ | $t/t_c = 0.33$ | $t/t_c = 0.67$ |
|---|---|---|---|
| $\varepsilon_{38}$ | 0.0842 | 0.0491 | 0.0747 |
| $\varepsilon_{75}$ | 0.0306 | 0.0170 | 0.0268 |
| $\varepsilon_{150}$ | 0.0090 | 0.0051 | 0.0081 |

ODEs (6) were integrated in time using an explicit, low-storage, 3-stage Runge Kutta scheme. The grid resolution employed to solve these PDEs was the same used for generating the velocity fields, $\Delta x/H = 1/150$, as in previous works [10, 11]. The interested reader can find the corresponding grid resolution study in the S5 Appendix. In the system of PDEs, uniform initial conditions were used for all variables, $u_i(\mathbf{x}, 0) = u_{i,0}, \overline{t_R}(\mathbf{x}, 0) = \overline{t_R^2}(\mathbf{x}, 0) = 0$, while for the ODE system the corresponding initial conditions were imposed, $u_i(0) = u_{i,0}$.

## Results

### Flow patterns and residence time

Pulsatile flow in the parent channel has two distinct phases coinciding with the acceleration and deceleration of the inflow profile prescribed at the inlet (Fig 2). The deceleration phase occurs for $0 \lesssim t/t_c \lesssim 0.5$, whereas the acceleration phase comprises the rest of the cycle. Fig 4 shows instantaneous vorticity fields (panels A–C) and streamlines (panels D–F) at three different instants of the cycle. At the onset of deceleration ($t = 0$, first column in Fig 4), the velocity is maximum in the parent channel and a counter-clockwise vortex is the dominant pattern inside the cavity. As deceleration proceeds ($t = 0.33t_c$, center column in Fig 4), the counter-

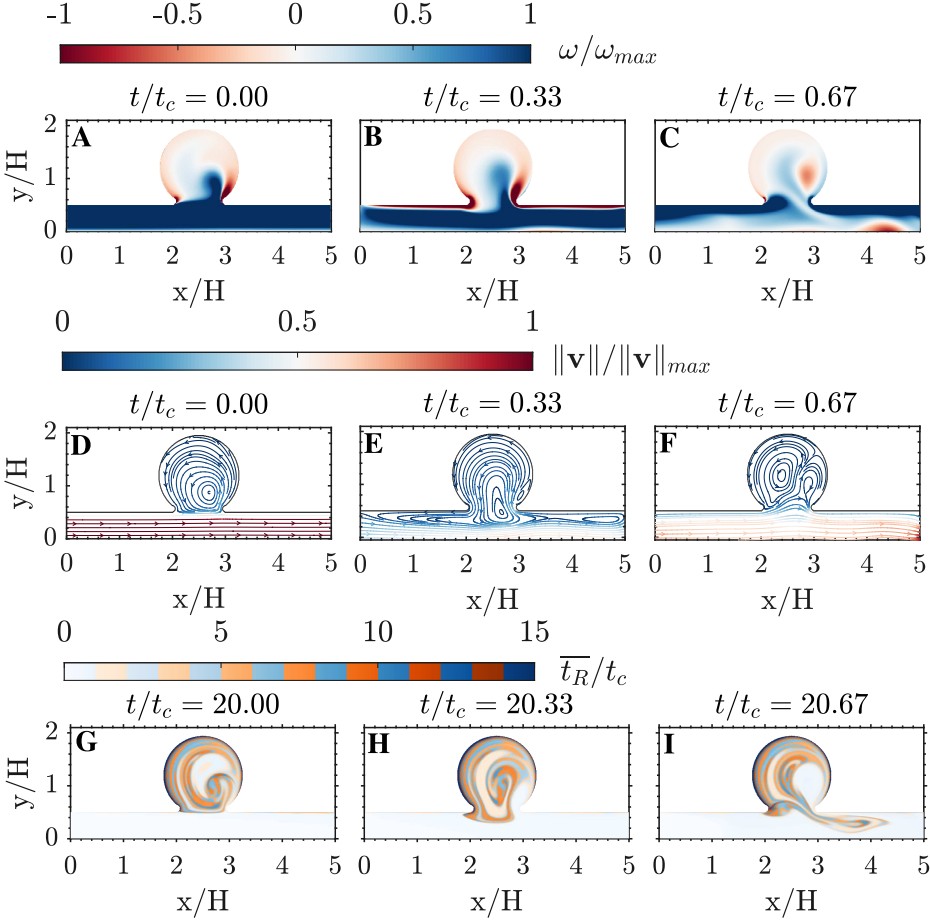

**Fig 4. Flow patterns in the cavity.** A-C: Vorticity and D-E: instantaneous streamlines colored by velocity magnitude, both normalized by its maximum value across the cardiac cycle. G-I: Residence time during the 21th simulation cycle. Each variable is plotted at three different phases of the cardiac cycle, as indicated on top of each panel.

clockwise vortex is partially sucked into the parent channel, creating a thin jet that drives fluid into the cavity in the downstream neck region while fluid slowly exits the cavity along the rest of the neck region. Finally, during acceleration ($t = 0.67t_c$, last column in Fig 4), a vortex pair appears inside the cavity and pulls fluid from the parent channel near the upstream neck region, ejecting fluid to the channel near the downstream neck region.

This alternating transport of fluid into and out of the cavity repeats every cycle, generating a layered structure in residence time values reminiscent of the growth rings in a tree trunk. Fig 4G, 4H and 4I highlight the developed layer pattern in the 21th cycle of the simulation, demonstrating that the WENO scheme effectively reproduces sharp $\overline{t_R}$ gradients over long periods of time without introducing excessive numerical diffusivity. However, albeit small, the WENO scheme does introduce a non-zero $D_n$ and, consequently, this scheme leads to $\sigma_T > 0$. This behavior is apparent in Fig 5, which displays snapshots of $\overline{t_R}$ and $\sigma_T$ over the period $10t_c \leq t \leq 21t_c$. Both variables grow with time, while keeping approximately the same spatial organization.

Fig 6 presents the temporal evolution of $\overline{t_R}$ and $\sigma_T$ at three points along the horizontal diameter of the cavity (i.e., the crosses in Fig 5A). In all three points, the residence time exhibits an initial phase of linear growth, $\overline{t_R} \approx t$ and $\sigma_T \approx 0$, as in our previous work [43]. At points that do not receive "fresh" fluid from the parent channel (e.g., the wall point in Fig 6A), this phase should last indefinitely. In our simulations, a small departure from $\overline{t_R} = t$ and $\sigma_T = 0$ becomes noticeable for $t \gtrsim 15t_c$ due to the WENO scheme's numerical diffusivity. Points exchanging fluid with the parent channel experience a different behavior characterized by two principal features. First, $\overline{t_R}$ rises and falls every cardiac cycle as pockets of stagnant and fresh fluid move back and forth over the point of interest. Second, the envelope of $\overline{t_R}$ saturates to a maximum value $\overline{t_{R\,max}}$ indicating the time needed for fluid exchange with the parent channel to wash out the local blood pool. At these points, $\sigma_T$ follows a similar behavior since fresh blood from the parent channel has not only low $\overline{t_R}$ but also low $\sigma_T$. We note that, for long enough simulation

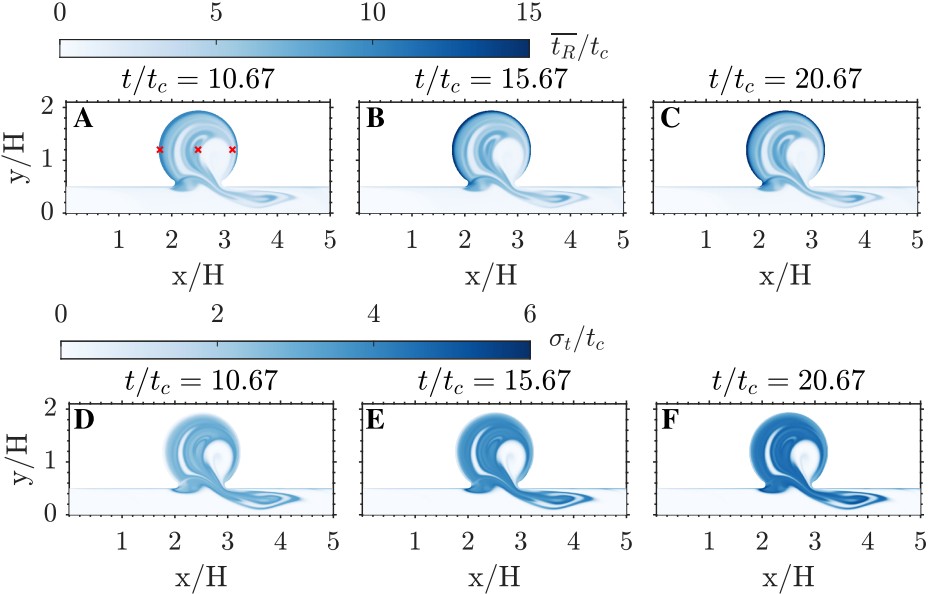

**Fig 5. Mean and standard deviation of the residence time.** A-C: spatial distribution of $\overline{t_R}/t_c$. D-F: spatial distribution of $\sigma_T/t_c$. Each variable is plotted at three different cycles. A,D: 11th cycle. B,E: 16th cycle. C,F: 21th cycle.

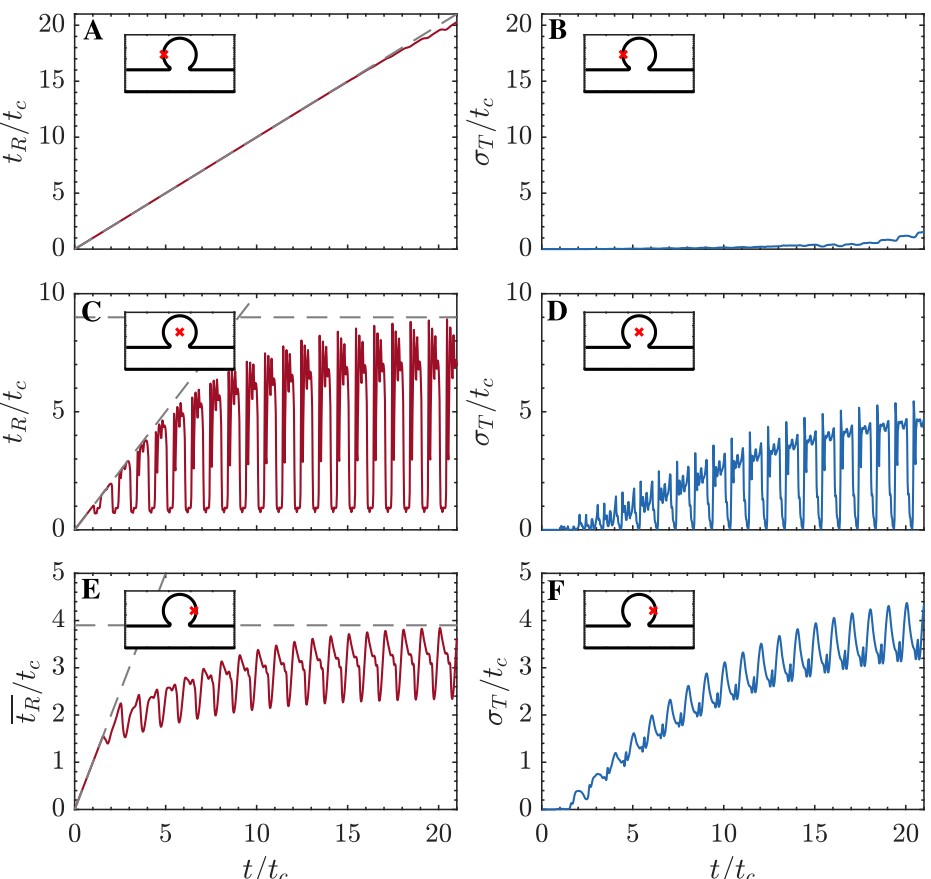

**Fig 6. Time series of residence time and its standard deviation.** A,C,E: Temporal evolution of $\overline{t_R}/t_c$ (—). B,D,F: Temporal evolution of $\sigma_T/t_c$ (—). Three locations are considered, indicated with × in Fig 5A: A,B at $(x/H, y/H) = (1.78, 1.19)$; C,D at $(x/H, y/H) = (2.5, 1.19)$; and E,F at $(x/H, y/H) = (3.15, 1.19)$.

times, $\sigma_T$ can grow at certain points to be comparable in magnitude to $\overline{t_R}$. This result has implications for multi-fidelity modeling of the coagulation cascade but also for the uncertainty of $\overline{t_R}$ itself.

## Multi-fidelity modeling of the coagulation cascade

The initiation of the coagulation cascade is characterized by the rapid growth of thrombin (IIa), peaking at about 15 cycles (see Fig 3). Fig 7 depicts the spatio-temporal structure of $u_{IIa}$ over the 16th simulation cycle, as obtained by the MuFi-1 (panels A–C) and MuFi-2 (panels D–F) models, as well as by the reference HiFi model (panels G–I). The three models capture the rise of $u_{IIa}$ inside the cavity and the formation of a layered structure similar to that of the residence time. There is a trend for MuFi-1 to underestimate $u_{IIa}$, which is for the most part corrected by MuFi-2.

To study each model's behavior in more detail, Fig 8 shows the thrombin concentration vs. time at the same three points considered in Fig 6, representative of scenarios where $\sigma_T \ll \overline{t_R}$, $\sigma_T \lesssim \overline{t_R}$, or $\sigma_T \gtrsim \overline{t_R}$ after running the simulation for a large number of cycles, $t \gg t_c$. For reference, the figure also shows the thrombin concentration obtained from the HiFi

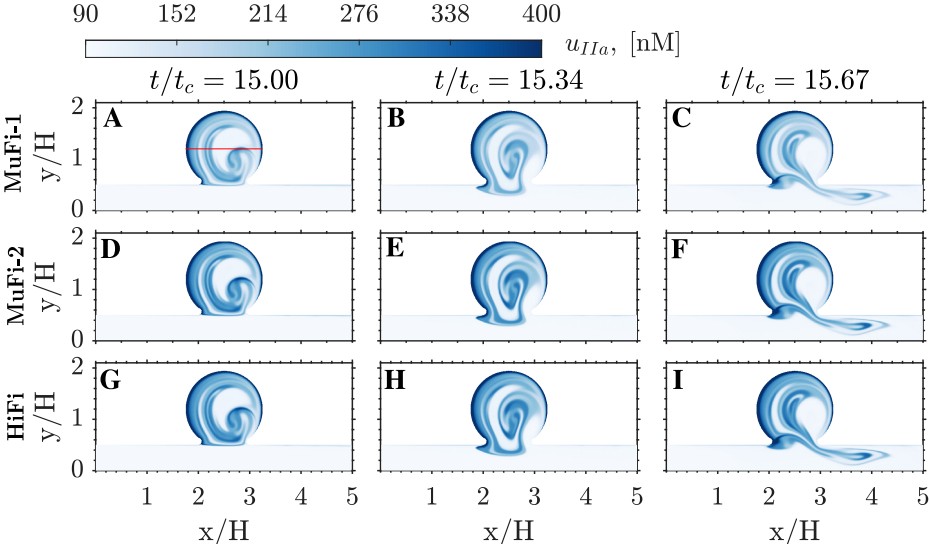

**Fig 7. Spatial distribution of thrombin concentration.** A-C: MuFi-1. D-F: MuFi-2. G-I: HiFi reference model. Three phases within the 16th cycle are plotted for each case, as indicated on the top row. A,D,G: $t/t_c = 15$. B,E,H: $t/t_c = 15.34$. C,F,I: $t/t_c = 15.67$.

model and the ODE system corresponding to no flow and no diffusion, i.e., $\overline{t_R} = t$ and $\sigma_T = 0$ (Eq 6).

Similar to the residence time, $u_{IIa}$ experiences peaks and valleys every cycle due to the periodic fluid exchange between the cavity and the parent channel. In addition, the peak value of $u_{IIa}$ increases from one cycle to the next as the coagulation cascade progresses in the fluid trapped in the cavity. The no-flow ODE model (dashed lines in Fig 8) fails to capture the oscillations of $u_{IIa}$ and severely overestimates the growth of its peak values. The MuFi-1 model captures the oscillatory nature of $u_{IIa}$, but it begins to underpredict the growth of its peak values after a number of cardiac cycles that varies from point to point. At the first sampled point, where $\sigma_T \ll \overline{t_R}$, the MuFi-1 model is almost exact for $t \lesssim 15 t_c$ and remains fairly accurate for $t \lesssim 20 t_c$ (Fig 8A). At the second and third sampled points, where $\sigma_T \sim \overline{t_R}$, the MuFi-1 model significantly departs from the HiFi model beyond $t \sim 10 t_c$ (Fig 8C and 8E). In contrast, the MuFi-2 model, which incorporates both $\overline{t_R}$ and $\sigma_T$, remains in reasonable agreement with the HiFi model much longer than the MuFi-1 model, producing fairly accurate results for simulation times well over 10 cycles at the three sampled points (Fig 8B, 8D and 8F). By $t = 20 t_c$, the MuFi-1 model underestimates the peak $u_{IIa}$ by 4.8%, 21.5%, and 15% while the MuFi-2 model does so by 2.7%, 1%, and 4.1%, respectively, at the first, second, and third sampled points.

Next, we compare the spatio-temporal behavior of the high-fidelity and multi-fidelity models by representing in Fig 9 the concentrations of thrombin, prothrombin, and factor Xa (i.e., $u_{IIa}$, $u_{II}$, and $u_{Xa}$) along the horizontal cavity diameter depicted in Fig 7A, with $y/H = 1.19$. Spatial concentration profiles are plotted at time-points $t/t_c = 10, 15, 20$. Factor Xa displays the best agreement between HiFi and MuFi models, followed by the prothrombin (II) and thrombin (IIa). Of note, the MuFi-2 model replicates the HiFi behavior for all three species, capturing their complex spatial oscillations. In comparison, the accuracy of the MuFi-1 model deteriorates faster with simulation time, although this model still retains the qualitative spatial dependence of species concentration.

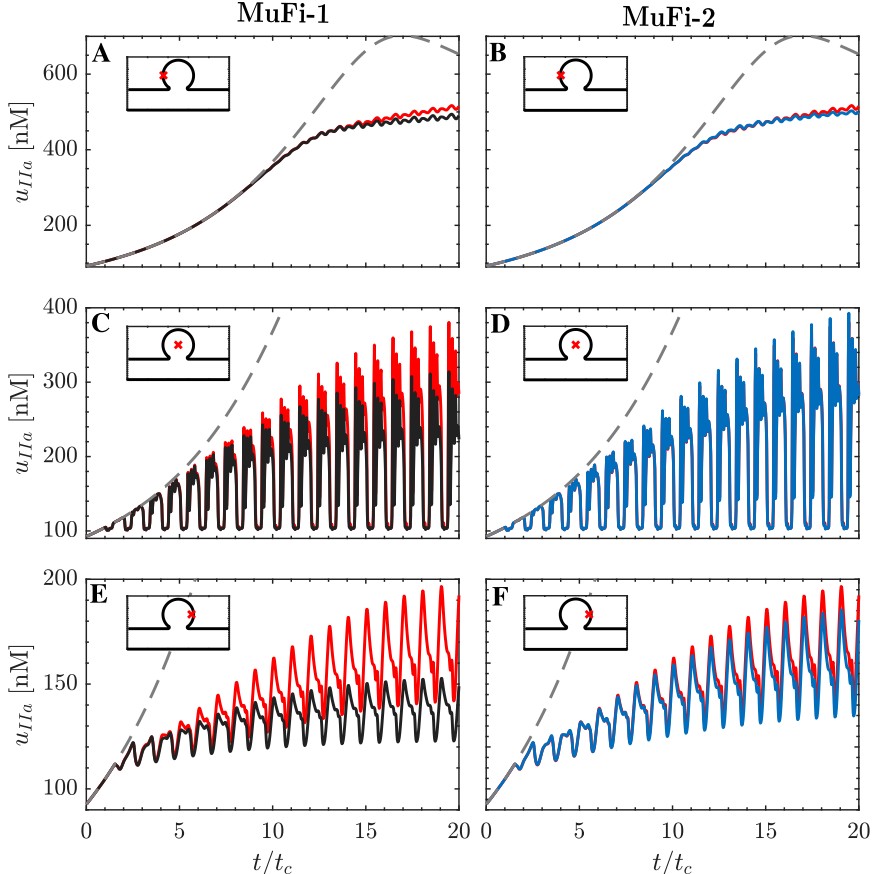

**Fig 8. Time series of thrombin concentration, $u_{IIa}$.** Each line correspons to a different model: MuFi-1 (—), MuFi-2 (—) and HiFi model (—). For reference, the solution of the 9-ODE system Eq (6) is also included (⋯). Three locations are considered, indicated with × in Fig 5A: A,B at $(x/H, y/H) = (1.78, 1.19)$; C,D at $(x/H, y/H) = (2.5, 1.19)$; and E,F at $(x/H, y/H) = (3.15, 1.19)$.

## Multi-fidelity model error analysis

While interesting to understand the performance of the multi-fidelity models, Figs 8 and 9 include examples of extreme behavior that do not reflect these models' overall accuracy. To systematically quantify the discrepancies between the high-fidelity and multi-fidelity models, we compute the relative errors

$$\varepsilon_i^p(x, y, t) = \frac{\left|u_i^{HiFi}(x, y, t) - u_i^{MuFi-p}(x, y, t)\right|}{u_i^{HiFi}(x, y, t)}, \tag{16}$$

where $i$ stands for species and $p$ for MuFi order. Fig 10 shows the thrombin relative errors of the MuFi-1 and MuFi-2 models, $\varepsilon_{IIa}^1$ and $\varepsilon_{IIa}^2$, for three instants along the 16th simulation cycle. These errors are negligible in the parent channel but reach appreciable values inside the cavity, where they exhibit a layered pattern with strong gradients, similar to $\overline{t_R}$ and $\sigma_T$. Also, $\varepsilon_{IIa}^1$ is significantly higher than $\varepsilon_{IIa}^2$, consonant with the data shown in Figs 7–9.

To characterize the dependence of the MuFi errors on the residence time and its variance inside the cavity, we divide the $(\overline{t_R}, \sigma_T)$ plane in bins, ensemble-average $\varepsilon_i^p$ inside each bin, and plot the resulting error maps in Fig 11 together with the corresponding normalized bin counts

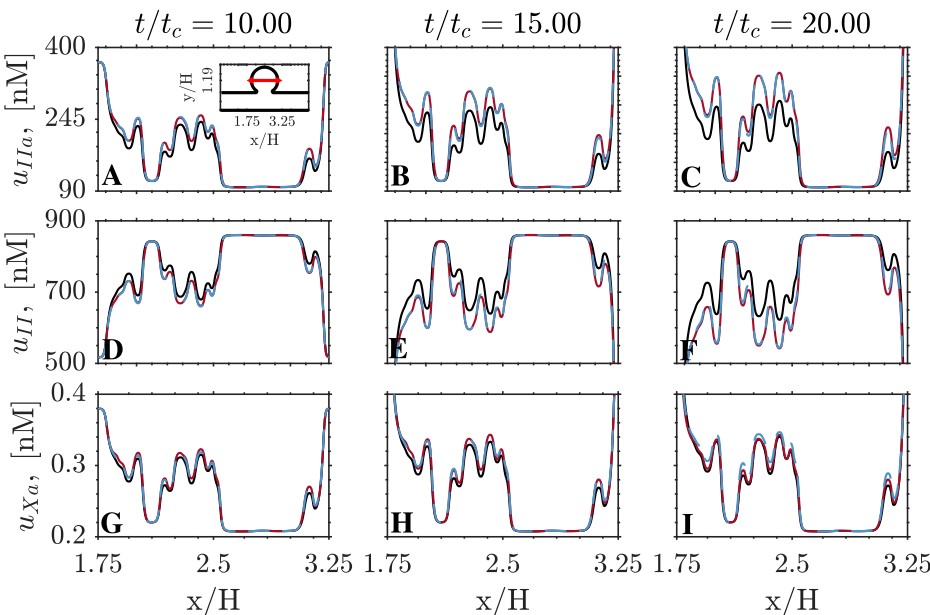

**Fig 9. Spatial distribution of species concentration.** Data is show at $y/H$ = 1.19 for MuFi-1 (——), MuFi-2 (——) and HiFi model (——). A,B,C: thrombin, $u_{IIa}$. D,E,F: prothrombin, $u_{II}$. G,H,I: factor Xa, $u_{Xa}$. Three different times are plotted for each species, as indicated on the top row.

[i.e., the probability density function $p(\overline{t_R}, \sigma_T)$]. The error maps obtained for different values of $t/t_c$ indicate that the MuFi error increases with $\sigma_T$ while it is much less sensitive to $\overline{t_R}$. The MuFi-2 model particularly outperforms the MuFi-1 model in areas of large $\sigma_T$, since MuFi-1 assumes $\sigma_T = 0$. Similar observations can be made for the error maps for factor Xa and prothrombin (II) concentrations, which are provided in S6 Appendix.

Inspection of the probability density functions shows that the majority of the points inside the cavity are circumscribed to two regions in the $(\overline{t_R}, \sigma_T)$ plane. The region near the point $(\overline{t_R}, \sigma_T) = (0, 0)$ corresponds to locations within the cavity that receive periodic inflows of

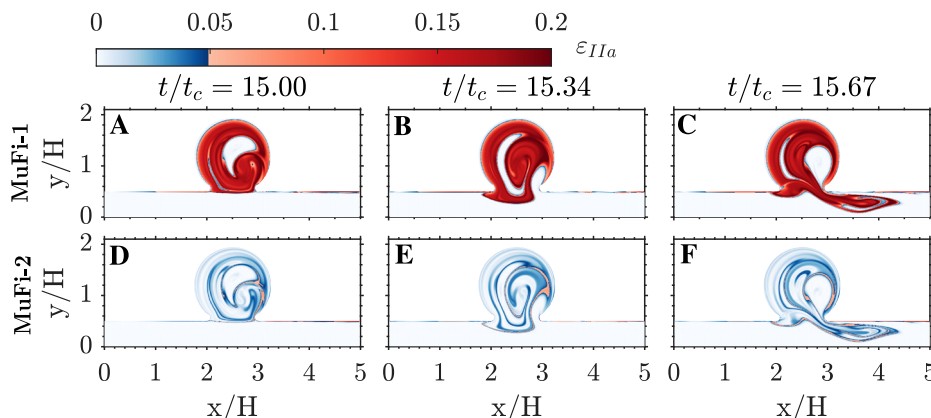

**Fig 10. Spatial distribution of relative errors in thrombin concentration.** The relative error $\varepsilon_{II_a}^p$ is defined in Eq (16). A-C: MuFi-1. D-F: MuFi-2. Three phases within the 16th cycle are plotted for each case, as indicated on the top row. A, D: $t/t_c$ = 15. B,E: $t/t_c$ = 15.34. C,F: $t/t_c$ = 15.67.

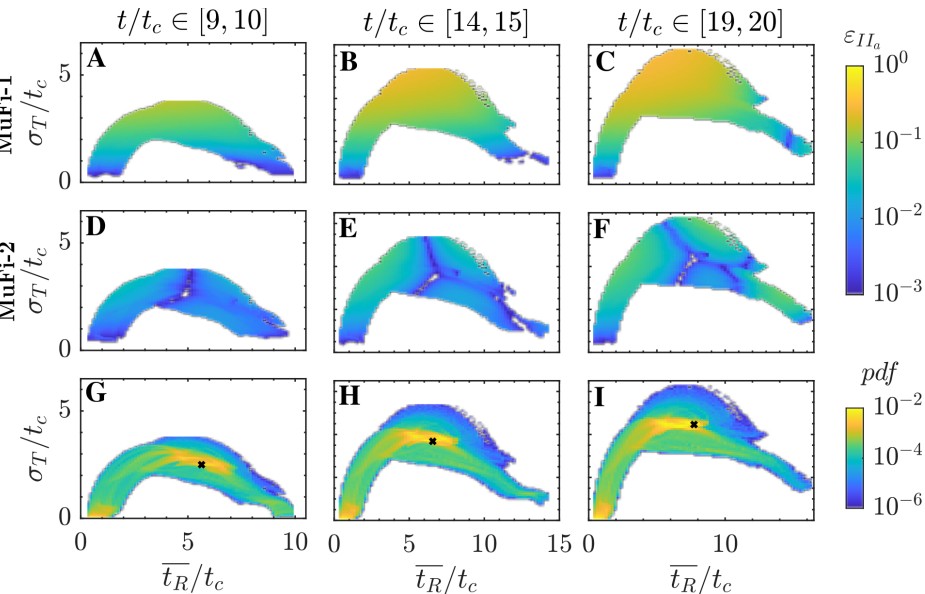

**Fig 11. Error maps for thrombin.** A-C: Relative error in thrombin concentration in MuFi-1, $\varepsilon_{II_a}^1$, as a function of residence time and its standard deviation. D-F: Same, but for MuFi-2, $\varepsilon_{II_a}^2$. G-I: Joint probability density function of residence time ($\overline{t_R}$) and its standard deviation ($\sigma_T$). Data for all panels is compiled inside cavity during three different cycles, as indicated on the top row. A,D,G: 10th cycle. B,E,H: 15th cycle. C,F,I: 20th cycle.

fresh flow from the parent channel. This periodic infusion sustains a low level of $\varepsilon_i^p$ throughout each cycle, since fresh flow has small values of $\overline{t_R}$ and $\sigma_T^2$. The region near $(\overline{t_R}, \sigma_T) = (5-10, 3-4)t_c$ corresponds to the stagnant areas of recirculating flow inside the cavity (see Fig 5). The modal errors in this region (black crosses in Fig 11G–11I) are 0.065, 0.141, and 0.187 for MuFi-1 and 0.004, 0.002, and 0.002 for MuFi-2 at $t/t_c$ = 9.5, 14.5, and 19.5, respectively. These stagnant flow areas displace upwards and towards the right in the $(\overline{t_R}, \sigma_T)$ plane as the simulation time advances (see bottom row of Fig 11). Consequently, we expect $\sigma_T$ and $\varepsilon_i^p$ to grow with $t/t_c$ inside the cavity and the MuFi-2 model to outperform the MuFi-1 model.

To test this hypothesis and quantify the overall error of MuFi models, we calculate the spatial average of $\varepsilon_i^p$ over the cavity region,

$$\overline{\varepsilon}_i^p(t) = \frac{1}{\Omega_{cav}} \int \int_{\Omega_{cav}} \varepsilon_i^p d\Omega, \tag{17}$$

and plot it vs. simulation time in Fig 12 for all 9 species. The results show differences among species but, overall, the MuFi-1 error $\overline{\varepsilon}_i^1$ grows sharply with time for $t \lesssim 10t_c$, then more gradually for longer times, even plateauing or tapering off in some cases. For most species, the MuFi-2 error $\overline{\varepsilon}_i^2$ is significantly lower than $\overline{\varepsilon}_i^1$ over extended periods of time or both errors are very low. For example, after 20 integration cycles, MuFi-2 is about an order of magnitude more accurate than MuFi-1 for thrombin and pro-thrombin, i.e., $\overline{\varepsilon}_{IIa}^1 = 0.14$ vs. $\overline{\varepsilon}_{IIa}^2 = 0.02$, and $\overline{\varepsilon}_{II}^1 = 0.08$ vs. $\overline{\varepsilon}_{II}^2 = 0.01$, whereas both models yield similar low errors for Factor XIa, i.e., $\overline{\varepsilon}_{Xa}^1, \overline{\varepsilon}_{Xa}^2 < 2 \times 10^{-3}$. The species showing the highest relative errors is PCa, reaching $\overline{\varepsilon}_{PCa}^1 = 0.8$ and $\overline{\varepsilon}_{PCa}^2 = 0.24$ at $t = 20t_c$. However, both the MuFi-1 error at $t = 10t_c$ and the MuFi-2 error at $t = 20t_c$ were well under 0.1 for all other species.

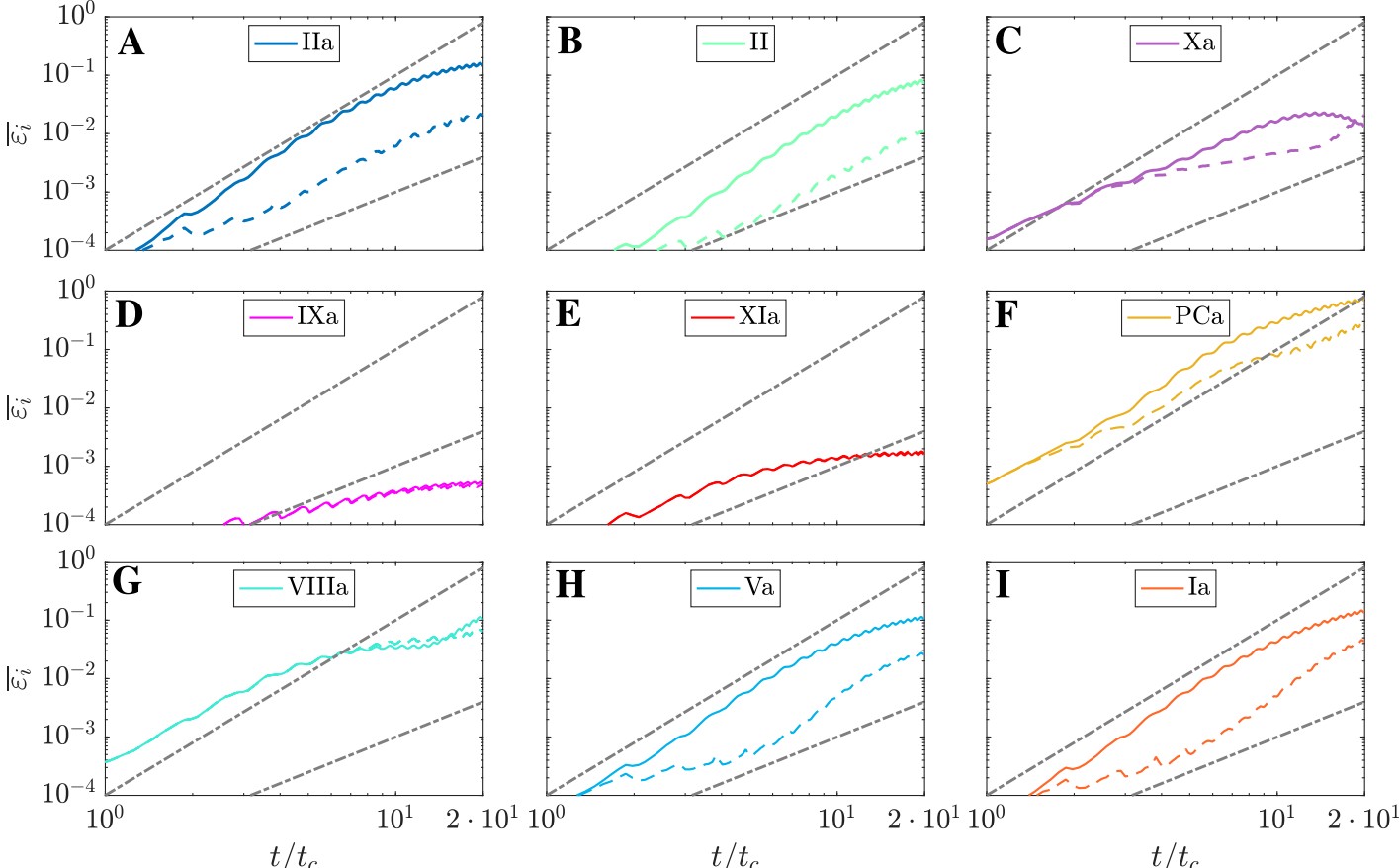

**Fig 12. Averaged relative error in the cavity.** Each line correspons to a different model: MuFi-1 solid, MuFi-2 dashed. Dashed-dot lines correspond to $\overline{\varepsilon}_i \propto (t/t_c)^2$ and $\overline{\varepsilon}_i \propto (t/t_c)^3$. A: Thrombin (IIa) B: Prohrombin (II) C: Factor Xa D: Factor IXa E: Factor Xa F: Activated protein C (PCa) G: Factor VIIIa H: Factor Va I: Fibrin (Ia).

To assess the dependence of the MuFi errors on coagulation model complexity, we repeated the numerical experiments described above using a 3-species system considering thrombin, factor XIa, and the inhibitor PCa [23]. This simplified system can be derived from the 9-equation system under the assumption that the faster chemical equations are in equilibrium. Overall, the performance of MuFi models was similar on the 3-equation and 9-equation systems (see S7 Appendix). For thrombin, the MuFi-1 and MuFi-2 models yielded respectively $\overline{\varepsilon}^1 = 0.16$ and $\overline{\varepsilon}^2 = 0.05$ at $t = 20t_c$ in the 3-equation system, comparable to the 9-equation values $\overline{\varepsilon}^1_{IIa} = 0.14$ and $\overline{\varepsilon}^2_{IIa} = 0.05$.

## Computational cost

The computational cost of solving 20 cycles of the 9-equation coagulation system using the HiFi approach was approximately 66.67 hours, using the MATLAB codes provided in [49] on an Intel(R) Xeon(R) Silver 4208 CPU @ 2.10GHz. On the same hardware, the MuFi-1 approach required around 4.24 hours, while the MuFi-2 approach took approximately 7.36 hours. This results in a speedup of 15.7 for MuFi-1, and 9.1 for MuFi-1, compared to the HiFi model. When the speedup estimated in Computational cost estimates taken into account (i.e.,

speedup $\approx (\varphi/\theta)(N/p)$), these numbers imply that the overhead associated to computing the reaction terms $R_i$ in HiFi (instead of the forcing terms for $\overline{t_R}$ and $\overline{t_R^2}$) is $\varphi/\theta \approx 2$.

## Discussion

The computational modeling of the coagulation cascade in flowing blood poses significant challenges due to the multi-scale nature of the process and the large number of involved chemical species. High-fidelity (HiFi) continuum mechanics models of coagulation lead to dozens of reaction-advection-diffusion partial differential equations (PDEs) [50]. The reaction kinetics in these equations are much slower than the cardiac cycle, requiring long simulation times to cover the coagulation process [9, 20, 22]. Moreover, the diffusion of chemical species is orders of magnitude slower than their convection and reaction kinetics, causing sharp scalar fronts that require ultra-high-resolution spatial meshes [25]. Despite the vast advances in computational software and hardware achieved in past decades, these joint requirements still impede the fast simulations of chemo-fluidic coagulation models in arterial or intracardiac domains, hindering the adoption of these models in clinical decision-making. Modelers usually resort to surrogate metrics associated with thrombogenesis, derived from blood residence time or wall shear stress [33, 51–55]. On the other hand, chemo-fluidic models of thrombosis are rare [10, 11] and often suffer from excessive diffusivity (numerical or explicit), short simulation times, and/or simplified coagulation models with few species.

We introduce a family of tailorable multi-fidelity (MuFi) models to effectively decouple the computational cost of simulating the coagulation cascade under flow from the number of reacting species, $N$. The MuFi models are designed to approximate the HiFi model in the limit of vanishing molecular diffusivity. In this limit, the $N$-PDEs of the HiFi model can be transformed into ordinary differential equations (ODEs) by changing variables between simulation time and blood residence time. Consequently, the HiFi model is replaced by $N$ ODEs representing the reaction kinetics and $p$ PDEs representing the ensemble mean residence time within each fluid particle, $\overline{t_R}$, and $p - 1$ higher-order statistical moments (namely, $\overline{t_R^2}, \overline{t_R^3}, \dots \overline{t_R^p}$) [56]. We provide a procedure to incrementally derive MuFi models of arbitrary order starting from the first-order model obtained with $p = 1$, corresponding to zero diffusivity. In the presence of small diffusivity, natural or numerical, higher-order models can be derived by Taylor-expanding the HiFi model around the zero diffusivity limit.

We assess the performance of the MuFi-1 and MuFi-2 models obtained for $p = 1, 2$ in a well-characterized, simplified coagulation system representing nine species: prothrombin (II), thrombin (IIa), fibrin (Ia), PCa, and factors XIa, IXa, Xa, VIIIa and Va [23]. This coagulation system is evaluated in a pulsatile flow through a two-dimensional geometry consisting of a parent channel driven by a Womersley inflow profile with a laterally protruding cavity where blood becomes stagnant. The non-dimensional parameters governing the flow, i.e., the Reynolds ($Re = 500$) and Womersley ($\alpha = 10$) numbers, are representative of a saccular aneurysm in an intermediate-size artery of the adult human circulatory system [57], although the instantaneous flow rate driving the flow does not include all the temporal complexity of physiological waveforms. For the residence time and its variance, the Peclet number is nominally set to be infinitely high by not including mass diffusivity terms in the corresponding PDE equations.

This configuration creates a cyclic influx of fresh fluid from the parent channel to the cavity and efflux of stagnant fluid from the cavity to the channel, together with a swaying fluid motion inside the cavity. Flow transport leads to a complex residence time pattern with thin layers separated by strong gradients, which intensifies as $\overline{t_R}$ grows with simulation time. Our WENO scheme resolve the thinly layered $\overline{t_R}$ pattern for long simulation times. However, albeit low, the WENO scheme has a non-zero numerical diffusivity [48], and thus, the effective $Pe$ in

our HiFi simulations is finite, affecting the solution to the transport equations for $\overline{t_R}$ and $\overline{t_R^2}$. Consequently, not only $\overline{t_R}$ but also its standard deviation, $\sigma_T$, increases with simulation time.

Overall, the MuFi-1 model compares well with the HiFi model, producing spatially averaged errors for thrombin inside the cavity that remain below 10% for up to 10 simulation cycles even if this model was derived under the assumption of $Pe \to \infty$ and $\sigma_T = 0$. Nevertheless, MuFi-1 starts to underestimate the concentration of all species beyond that point. By $t = 20t_c$, the spatially averaged errors for thrombin inside the cavity are as high as $\approx$20%. The MuFi-2 model yields significantly lower errors than the MuFi-1 model, with spatially averaged errors for thrombin that remain below 2% for up to $t \approx 20t_c$. These errors are comparable to those associated to solving the HiFi model's PDEs (see S5 Appendix). The accuracy of the MuFi models is worst for PCa, the species with the fastest growth rate in the considered coagulation model. Statistical analysis shows that the MuFi errors depend almost exclusively on $\sigma_T$ and that this dependence is steeper for MuFi-1 than for MuFi-2.

Because $\sigma_T$ increases with numerical diffusivity, $D_n$, the performance of MuFi implementations is expected to depend on the numerical scheme and the spatio-temporal resolution used to discretize the model PDEs. While an exhaustive analysis of the numerical diffusivity of numerical discretizations of advection-reaction-diffusion problems is beyond the scope of this study, we have outlined a general methodology to formulate MuFi models of arbitrary order, derive each numerical scheme's EDE for $\sigma_T$, and outlined a step-by-step process to compute $\sigma_T$ without the need to derive its EDE. With these tools, it should be straightforward to tailor MuFi models to the peculiarities of each flow and numerical solver. These step-by-step procedures should also apply to models that compute residence time using non-classical numerical representations of the governing PDEs such as, e.g., neural networks.

The MuFi models ($N$ ODEs, $p$ PDEs) are much more efficient than the HiFi model ($N$ PDEs) because solving ODEs is significantly cheaper than solving PDEs. In physiologically relevant computational meshes containing hundreds of elements in each direction, MuFi models achieve a speedup $> N/p$. We anticipate this speedup to exceed an order of magnitude, considering the favorable accuracy achieved by low-order MuFi models ($p = 1, 2$) and the dozens of species involved in realistic coagulation cascade models. An alternate interpretation is that the computational cost of MuFi models becomes independent of the number of species $N$. Additionally, one can run any number of different MuFi models inexpensively for a specified flow baseline. The costly part of MuFi models, i.e., solving the p-PDEs representing residence time and its higher-order moments, does not need to be redone unless the anatomy or inflow/outflow conditions change. Thus, MuFi models are highly efficient for sensitivity analyses, uncertainty quantification, kinetics model comparisons, and any type of study requiring multiple evaluations of the coagulation cascade model.

In the coagulation cascade, the diffusive and chemical timescales are, overall, significantly different, justifying the idea of MuFi models. However, the non-linearity of the reaction rate terms makes it difficult to rule out that diffusion became dominant for some species in some regions of state space, especially as the number of species increases. While this situation could make it necessary to increase the MuFi order as $N$ increases, deteriorating the MuFi speedup, we have seen no indication of it being significant. Specifically, MuFi models reproduced the HiFi results for thrombin with similar accuracy on a classic 9-equation model [23], and its 3-equation simplified version. Investigation of MuFi models with $N > 9$ should clarify this matter further.

For simplicity and to establish proof of concept of the MuFi strategy, this study focuses on the intrinsic coagulation cascade, ignoring the extrinsic cascade initiated by the release of pro-coagulatory or inhibitory species from the vessel walls. While both the intrinsic and extrinsic

coagulation pathways are often activated in cardiovascular conditions associated with thrombosis [58–60], focusing on the intrinsic pathway is not uncommon in the literature, motivated by the paucity of information about the wall's prothrombotic potential in many clinically relevant scenarios [15]. Generating MuFi models for the extrinsic pathway would involve additional PDEs representing the residence time of wall-released chemical species or the time spent by blood near damaged wall regions, e.g., the near-wall residence time proposed by others [11]. As long as the number of additional PDEs remains lower than the total number of species, $N$, the resulting MuFi models would still save computational time. Furthermore, applications requiring multiple HiFi runs, e.g., uncertainty quantification studies, simulations pharmacological treatments, etc., can still be accelerated by MuFi models, even if this required solving a significant number of residence-time-like PDEs. A notable exception would be, however, applications requiring two-way coupling between flow and coagulation.

Finally, note that residence time and its higher order moments can not only be obtained *in silico* using CFD analysis [43, 44, 51], but also *in vitro* using experimental techniques like particle image velocimetry [61–63]. *In vitro* experiments offer opportunities to validate HiFi and MuFi approaches by, e.g., tracking the trajectories of single particle tracers or the evolution of dye injections. Moreover, residence time can also be estimated *in vivo* using medical imaging modalities like phase-contrast magnetic resonance imaging, Doppler ultrasound, and perfusion computed tomography [64–67]. Residence time maps obtained *in vivo* could be fed to the ODE component of the MuFi models, providing a computationally tractable method to calculate the spatio-temporal evolution of the coagulation species non-invasively in the clinical setting.

## Conclusion

We present a novel multi-fidelity approach to mitigate the computational burden of simulating the coagulation cascade under flow. Using residence time $\overline{t_R}$ and its statistical moments, the multi-fidelity (MuFi) modeling achieves a favorable trade-off between computational cost and accuracy. The multi-fidelity approach allows for the integration of various data streams, including CFD analysis, *in vitro* experiments, and non-invasive imaging *in vivo* techniques, enabling a comprehensive understanding of the spatial and temporal progression of coagulation species and offering promise for clinical translation. Finally, we note that the same multi-fidelity strategy developed here can be adopted to increase the efficiency of simulating other systems biology processes influenced by blood flow.

## Supporting information

**S1 Appendix. Itô's differentiation.**
(PDF)

**S2 Appendix. Multi-fidelity model of third order.**
(PDF)

**S3 Appendix. Womersley inflow boundary condition.**
(PDF)

**S4 Appendix. Coagulation cascade model.**
(PDF)

**S5 Appendix. Grid resolution study for the HiFi model.**
(PDF)

**S6 Appendix. Error maps for prothrombin and factor Xa.**
(PDF)

**S7 Appendix. Averaged relative errors for $N = 3$.**
(PDF)

## Author Contributions

**Conceptualization:** Manuel Garcia-Villalba, Juan C. del Alamo, Oscar Flores.

**Data curation:** Alejandro Gonzalo.

**Formal analysis:** Manuel Guerrero-Hurtado.

**Funding acquisition:** Manuel Garcia-Villalba, Pablo Martinez-Legazpi, Andrew M. Kahn, Elliot McVeigh, Javier Bermejo, Juan C. del Alamo, Oscar Flores.

**Investigation:** Manuel Guerrero-Hurtado.

**Software:** Manuel Guerrero-Hurtado, Alejandro Gonzalo, Oscar Flores.

**Supervision:** Manuel Garcia-Villalba, Andrew M. Kahn, Elliot McVeigh, Javier Bermejo, Juan C. del Alamo, Oscar Flores.

**Writing – original draft:** Manuel Guerrero-Hurtado, Manuel Garcia-Villalba, Juan C. del Alamo, Oscar Flores.

**Writing – review & editing:** Manuel Guerrero-Hurtado, Manuel Garcia-Villalba, Alejandro Gonzalo, Pablo Martinez-Legazpi, Juan C. del Alamo, Oscar Flores.

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
