## [Decision Letter · Decision Letter 0]

3 Jul 2023

Dear Dr Flores,

Thank you very much for submitting your manuscript "Efficient multi-fidelity computation of blood coagulation under flow" for consideration at PLOS Computational Biology.

As with all papers reviewed by the journal, your manuscript was reviewed by members of the editorial board and by several independent reviewers. In light of the reviews (below this email), we would like to invite the resubmission of a significantly-revised version that takes into account the reviewers' comments.

We cannot make any decision about publication until we have seen the revised manuscript and your response to the reviewers' comments. Your revised manuscript is also likely to be sent to reviewers for further evaluation.

Sincerely,

Alison L. Marsden

Academic Editor

PLOS Computational Biology

Pedro Mendes

Section Editor

PLOS Computational Biology

Reviewer's Responses to Questions

**Comments to the Authors:**

Reviewer #1: In this article, the authors propose a method to reduce the cost of thrombosis modeling. They achieve this cost reduction by reducing the number of PDEs that one has to typically solve to resolve the concentration of various species. In their proposed technique, they only solve for the residence time's PDE (and perhaps an additional PDE) and compute the concentration of various species based on the residence time through a set of ODEs that are cheap to solve.

The idea is compelling, the study is designed properly, and the manuscript is well-written. Nevertheless, there are major limitations associated with this technique. While the authors acknowledge some in the text, they are not as explicit about others. Those must be clearly mentioned in the abstract/discussion/conclusion.

First and foremost, the theoretical estimate of the cost reduction is tied to error. It is true that there is more cost saving if there are more PDEs to solve to begin with. But I am guessing the error in these computations will also be larger as N increases. To show that, note that you are effectively trying to solve $\\dot Y = RY + LY$ where the vector $Y$ contains all discrete solutions, matrix $R$ is the linearized operator associated with reaction terms, and matrix $L$ is the discrete Laplace operator. The solution to this problem is expressed in terms of eigenvalues of $R+L$. The accuracy of your approximation depends on how close eigenvalues of $R$ are to those of $R+L$. It is true that as $||L||$ goes to zero (due to low diffusivity), the approximation becomes more accurate. But I would argue that as the number of species goes up (i.e., N), then there will be a wider range of eigenvalues in $R$ that could get manipulated depending on whether $L$ is dropped or not. Physically speaking that corresponds to kinetics that occur slowly (for instance the coupling associated with two species unfolds slowly), thus allowing for the diffusion process to kick in. Another supporting evidence for this argument is that the effect of modified eigenvalues on the solution accuracy grows with integration time, thus lowering the accuracy of the proposed approach with time. The reported results confirm this as they show an increase in error with time. To summarize either the authors must either acknowledge that the reported errors will likely go up as N increases or present results with an N>>3 that shows the counter.

The second major point is that the authors never mention the actual cost reduction. Even though that is the selling point of the proposed method, they do not report how much cost reduction they obtained in reality. I understand that some people argue that it depends on the implementation etc ... but here the authors are using identical implementations for solving residence time PDE as those of the species. Hence, the cost numbers will provide an apple-to-apple comparison for the baseline and proposed methods. On the same note, the authors should mention what was $N$ that they considered in the abstract. Reporting other values without mentioning N is meaningless.

The two other limitations of the study that must be mentioned, perhaps in a dedicated section are 1) the fact that BCs for all species must be identical (there is some discussion of that in the manuscript), 2) this is only suitable for cases where thrombosis modeling is one-way coupled to the Navier Stokes. Namely, if the viscosity or boundary conditions get modified due to the thrombosis, then one must solve kinetic equations coupled with those of the fluids. Hence, this method will not be practical for those settings.

Two minor comments:

1) line 26: did they mean tissue factor?

2) computation of residence time dates much earlier than reference 29. See for example Józsa, János, and Tamás Krámer. "Modelling residence time as advection-diffusion with zero-order reaction kinetics." Proceedings of the Hydrodynamics 2000 Conference, International Association of Hydraulic Engineering and Research. 2000. or Esmaily-Moghadam, Mahdi, Tain-Yen Hsia, and Alison L. Marsden. "A non-discrete method for computation of residence time in fluid mechanics simulations." Physics of fluids 25.11 (2013): 110802.

Reviewer #2: This paper deals with the challenges associated in solving dozens number of CDR equations to model the coagulation cascade in flowing blood. Besides the computational time, and due to large Pe number, it is usually achieved by adding numerical resolution such as upwinding. Therefore, this work proposes a very nice contribution by introduce a tailorable multi-fidelity (MuFi) models to reduce the computational cost associated with the simulating of the coagulation cascade of N species in a flow. It transforms N-PDEs of the high fidelity resolution into one or two PDEs with N ordinary differential equations by changing variables between simulation time and blood residence time.

The new multi-fidelity framework is tested on 2D idealized aneurysm and the comparisons between the models are discussed. Despite several assumptions mainly related to diffusivity, the framework presents a very nice contribution to the understanding of the spatial and temporal progression of coagulation species. For all that reasons, the reviewer recommends publication while answering the following questions and adding them to the revised version:

• Does the resolution of the PDE in MuFi uses the same scheme as the N PDE Hifi?

• Can the authors comment on the extension 3D simulation? is it straightforward?

• Has the initial HiFi resolution been validated before its use?

• The authors must provide a figure on the used mesh for the resolution of NS, and the PDEs.

• May the authors provide a summary table on the cost of each method to well compare and appreciate the contribution of the Mufi approaches?

• The resolution of NS and the PDEs are decoupled, which may affect the blood viscosity, can the authors provides some comments on this point?

Reviewer #3: The authors presented a new numerical approach for the computation of species concentration as a function of blood residence time. Their ultimate goal is to use this approach to model the coagulation cascade. The work is of high interest as the proposed approach could significantly reduce the computation time while exhibiting a rather low error.

However, there are a few points that I would like to highlight.

Introduction

Line 18. The authors list here many computational models for coagulation. However, as they mention earlier in the introduction, coagulation can be triggered by many factors (abnormal blood flow, hypercoagulability and vessel injury). Could the authors please provide more details about how the different models they refer to articulate around these 3 factors potentially leading to thrombus formation?

Line 27. As the authors mentioned, coagulation involves different time scales. Could the authors please provide a short summary of the different time scales involved? It would really help to appreciate the challenges of such numerical modelling.

Methods

Line 130. The authors explain that the simplified model has no mixing, and that the reaction rate only depends on its age, which allows them to focus on residence time. Is this a strong hypothesis? If it is a common hypothesis, could the authors please refer to the appropriate literature?

Line 146. The authors stated that MuFi models can be significantly cheaper than the HiFi model. Could they please provide an estimate of this “cost” difference?

Line 239. The authors used a 2D idealized geometry for their computation. Could the authors comment on their choice? Does this geometry usually result in thrombosis? They also mention the left atrial appendage as an application, but this anatomical feature exhibits high velocity gradient (high velocity at the inflow and very low velocities at the tip). Have the authors tested their modeling framework for flow fields exhibiting strong velocity gradients? Also, have they tested them in more complex flow topologies (like the left ventricle for instance)?

Line 283. The authors sampled the cycle with 35 points and then linearly interpolated. Have the authors tested different sampling rates? Could they comment on the size of the sampling interval compared to the phenomena they want to model?

Results

Line 309. The authors used a symmetric waveform, when a typical systole/diastole ratio would be around 0.3/0.7. Could the authors justify their assumption?

Line 342. The authors highlight the fact that the standard deviation of the residence time reaches values that can be comparable with the mean value. Could the authors please detail what are the implications of this large value of std for the multi-fidelity modeling?

Line 380. The special distribution of species concentrations showed variable levels of agreement between the different species. Could the authors provide a possible explanation for the differences they observed between the different species?

Line 381. The authors state that the MuFi-2 model accurately replicated the HiFi model up to 15 cardiac cycles. Could the authors comment on whether this time is enough to correctly capture the reactions involved in coagulation? More specifically could they comment on the ratio between this simulation time and the coagulation model parameters K?

Discussion

Line 437. The authors state that the reaction kinetics of these equations are much slower than the cardiac cycle. Could they please provide an estimation for this? It will help appreciate the value of their modelling approach.

Line 483. The average error for the MuFi-2 model is about 5% for 20 cardiac cycles. Have the authors investigated how will this error evolve as the number of cycles increases? More generally, would this model be suitable for simulation over longer time scales?

Line 531. The authors describe other approaches to obtain residence time, from in vivo or in vitro models. Have they thought about any way to confront their numerical approach with in vitro experimental data for instance?

Minor comments

Figure 3. Could the authors please add the species associated with u1-3 in the legend of the figure?

Figure 5. The color scale for the plot of the mean residence time tends to be hard to read. Could the authors please use a linear color scale?

Figure 6. Could the authors add a schematic for the locations of the 3 data points? They refer to Figure 5A, but having all the information in Figure 6 would help the reader. This comment can also apply to Fig 9.

Figures. Overall, the tables of figures are sometimes difficult to read. Having the type of model included in the figure would help.

Line 229. Please add a space before 14.

Line 412. Please add a space before the bracket.

**Have the authors made all data and (if applicable) computational code underlying the findings in their manuscript fully available?**

Reviewer #1: None

Reviewer #2: Yes

Reviewer #3: Yes

PLOS authors have the option to publish the peer review history of their article (what does this mean?). If published, this will include your full peer review and any attached files.

Reviewer #1: No

Reviewer #2: **Yes: **Prof. Elie Hachem

Reviewer #3: No
---

## [Decision Letter · Decision Letter 1]

9 Oct 2023

Dear Dr Flores,

We are pleased to inform you that your manuscript 'Efficient multi-fidelity computation of blood coagulation under flow' has been provisionally accepted for publication in PLOS Computational Biology.

Best regards,

Alison Marsden

Academic Editor

PLOS Computational Biology

Pedro Mendes

Section Editor

PLOS Computational Biology

Reviewer's Responses to Questions

**Comments to the Authors:**

Reviewer #1: No further comments. Accept

Reviewer #2: All the comments are answered in the revised version

Reviewer #3: The authors have addressed all my previous comments.

**Have the authors made all data and (if applicable) computational code underlying the findings in their manuscript fully available?**

Reviewer #1: None

Reviewer #2: None

Reviewer #3: None

PLOS authors have the option to publish the peer review history of their article (what does this mean?). If published, this will include your full peer review and any attached files.

Reviewer #1: No

Reviewer #2: No

Reviewer #3: No

---

## [Editor Report · Acceptance letter]

16 Oct 2023

PCOMPBIOL-D-23-00859R1 

Efficient multi-fidelity computation of blood coagulation under flow

Dear Dr Flores,

I am pleased to inform you that your manuscript has been formally accepted for publication in PLOS Computational Biology. Your manuscript is now with our production department and you will be notified of the publication date in due course.

With kind regards,

Zsofi Zombor
